# Ultrasound-Guided Interventions for Neuropathic Pain: A Narrative Pictorial Review

**DOI:** 10.3390/life15091404

**Published:** 2025-09-05

**Authors:** Ting-Yu Lin, Ke-Vin Chang, Wei-Ting Wu, Kamal Mezian, Vincenzo Ricci, Levent Özçakar

**Affiliations:** 1Department of Physical Medicine and Rehabilitation, Lo-Hsu Medical Foundation, Inc., Lotung Poh-Ai Hospital, Yilan 265501, Taiwan; t840326@icloud.com; 2Graduate Institute of Clinical Medicine, College of Medicine, Taipei Medical University, Taipei 110031, Taiwan; 3Department of Physical Medicine and Rehabilitation, National Taiwan University Hospital, College of Medicine, National Taiwan University, Taipei 100225, Taiwan; wwtaustin@yahoo.com.tw; 4Department of Physical Medicine and Rehabilitation, National Taiwan University Hospital, Bei-Hu Branch, Taipei 100229, Taiwan; 5Center for Regional Anesthesia and Pain Medicine, Wang-Fang Hospital, Taipei Medical University, Taipei 116081, Taiwan; 6Department of Rehabilitation Medicine, First Faculty of Medicine and General University Hospital in Prague, Charles University, 12000 Prague, Czech Republic; kamal.mezian@gmail.com; 7Physical and Rehabilitation Medicine Unit, Luigi Sacco University Hospital, ASST Fatebenefratelli-Sacco, 20157 Milano, Italy; vincenzo.ricci58@gmail.com; 8Department of Physical and Rehabilitation Medicine, Hacettepe University Medical School, Ankara 06100, Turkey; lozcakar@yahoo.com

**Keywords:** ultrasound, sonography, injection, chronic pain, nerve block

## Abstract

Neuropathic pain presents a persistent therapeutic challenge, arising from diverse etiologies such as trigeminal neuralgia, postherpetic neuralgia, post-amputation pain, painful polyneuropathy, peripheral nerve injury pain, and painful radiculopathy. Given the limitations and side effects associated with pharmacologic treatments, interest in interventional therapies has surged. Herein, ultrasound guidance provides real-time, radiation-free visualization that enhances procedural accuracy and safety. This narrative review synthesizes current evidence on ultrasound-guided techniques—including nerve blocks, pulsed radiofrequency, hydrodissection, and peripheral nerve stimulation—in the management of neuropathic pain. These minimally invasive approaches demonstrate potential in providing significant and durable pain relief, enhancing functional outcomes, and reducing reliance on systemic medications. Notably, much of the existing literature comprises small-scale or observational studies and larger randomized controlled trials are therefore essential to confirm efficacy, define optimal treatment parameters, and inform clinical guidelines for broader adoption.

## 1. Introduction

Neuropathic pain is pain resulting from a lesion or dysfunction within the somatosensory nervous system which extends from peripheral receptors located in the skin, muscles, and joints, to central pathways involving the spinal cord and higher cortical processing centers in the brain [1]. Under normal physiological conditions, pain serves as a protective mechanism, alerting the body to tissue damage. However, when there is disruption to the somatosensory pathway—whether due to trauma, infection, metabolic disorders, or neurodegenerative processes—aberrant signaling may occur [2,3,4]. Peripheral sensitization stems from alterations in nerve fiber density and ectopic hyperexcitability driven by changes in nerve membrane composition, synaptic properties, and neurotransmitter expression [3,5]. These faulty discharges are subsequently propagated from peripheral neurons to their central targets. Central sensitization arises from synaptic changes in second-order neurons, microglial hyperactivation, dysfunctional descending inhibition pathways, and maladaptive plasticity in subcortical and cortical regions [5,6,7]. This creates painful symptoms such as burning, shooting, or electric shock-like sensations, often accompanied by sensory abnormalities like allodynia or hyperalgesia. Based on anatomical classification, neuropathic pain can be categorized as central (e.g., stroke, spinal cord injury, multiple sclerosis, Parkinson’s disease) or peripheral in origin (e.g., diabetes, radiculopathy, chemotherapy-induced peripheral neuropathy, acute inflammatory demyelinating polyneuropathy) [3]. The diagnosis is based on careful history taking, physical examination, and confirmatory tests [2]. Various questionnaires have been developed to help distinguish between neuropathic and non-neuropathic pain by identifying discriminative clinical features [8]. Large population-based studies discovered that approximately 7–40% of chronic pain cases contain a neuropathic component, leading to great health burdens [9,10].

Pain control remains the primary goal of treatment for neuropathic pain, as the original culprit is usually difficult to eliminate. First-line pharmacologic therapies include pregabalin, gabapentin, duloxetine, and tricyclic antidepressants [11]. High-concentration capsaicin patches, lidocaine patches, and tramadol are second-line options [1]. Unfortunately, most medications for neuropathic pain demonstrate only moderate efficacy, and their clinical use is often limited by adverse effects. As a result, interventional therapies, e.g., nerve blocks, steroid injections, spinal cord and peripheral nerve or even deep brain stimulation are becoming increasingly popular.

Ultrasound is a widely utilized therapeutic and diagnostic tool. Therapeutic ultrasound has been shown to reduce neuropathic pain by modulating neurotransmission and inflammatory pathways, and it may even facilitate nerve regeneration [12,13]. In diagnostic applications, the intricate architecture of peripheral and truncal nerves could be tracked effectively by high-resolution ultrasound [14,15,16]. It is also useful for guiding injections, as it enables real-time visualization of the needle throughout the procedure without radiation exposure [17]. By allowing for clear delineation of surrounding structures, it enhances both precision and safety. Ultrasound-guided interventions represent a new approach to treating neuropathic pain. Given the diversity of intervention techniques and study designs in this broad, evolving field, we elected to conduct a narrative review. This review summarizes current evidence to provide an updated perspective on the efficacy of ultrasound-guided interventions in the management of neuropathic pain. Most previous reviews have either compiled interventions by various imaging modalities or concentrated exclusively on specific pain conditions. To our knowledge, this is the first review specifically dedicated to ultrasound-guided interventions spanning the full range of neuropathic pain etiologies.

## 2. Literature Search

A systematic literature search was performed across four electronic databases—PubMed, Web of Science, Embase, and the Cochrane Library—from their inception to 26 July 2025. The search strategy incorporated the following keywords: “ultrasound-guided”, “ultrasound”, “sonography”, “nerve”, “neuropathic pain”, “neuropathy”, and “neuralgia”. Studies were eligible for inclusion if they (1) involved human subjects and (2) evaluated or introduced the use of ultrasound-guided interventions for the treatment of neuropathic pain. Studies were excluded if they focused solely on diagnostic ultrasound applications or used animal models.

Previous reviews have classified neuropathic pain into central and peripheral types; with the latter category further divided into trigeminal neuralgia, postherpetic neuralgia, post-amputation pain, painful polyneuropathy, peripheral nerve injury pain, and painful radiculopathy [1,18]. Adopting this framework, we provide an overview of ultrasound-guided interventions, outlining the types of procedures, technical approaches, and clinical outcomes for each condition, accompanied by illustrations of relevant techniques.

## 3. Trigeminal Neuralgia

Trigeminal neuralgia presents as unilateral, paroxysmal facial pain in one or more divisions of the trigeminal nerve, often triggered by innocuous stimuli [19]. The lifetime prevalence is estimated as 0.16–0.3%, with a female predominance. The most common etiology is neurovascular compression at the root entry zone, where the nerve exits the brainstem and transitions to peripheral fibers. This produces focal irritation and demyelination [20]. First-line treatment is phrenological, whereby surgical options are reserved for those with retractable symptoms [21]. In patients unresponsive to medical therapy, an ultrasound-guided nerve block via the pterygopalatine fossa could relieve pain for up to 15 months [22].

Injection techniques to the various branches of the trigeminal nerve have been described [23]. Local anesthetics (e.g., 1–5 mL 0.5% lidocaine), could be delivered using a linear or curvilineal transducer, selected based on the depth of the nerve. Doppler imaging should be used routinely to avoid vascular structures prior to injection. The supraorbital nerve, a branch of the ophthalmic division (V1) of the trigeminal nerve, emerges from the supraorbital notch approximately 2–3 cm lateral to the midline (Figure 1A,B). With the patient’s head in neutral position, a linear transducer is placed transversely over the medial eyebrow. The supraorbital notch is recognized as an interruption in the hyperechoic contour of the superior orbital rim. A lateral-to-medial in-plane approach is used for nerve blockade. The infraorbital nerve, a terminal branch of the maxillary division (V2), can be targeted by aligning the transducer parallel to the body of the maxilla, approximately 1 cm inferior to the infraorbital margin (Figure 1A,C). The needle is advanced in-plane from lateral to medial toward the infraorbital foramen. To block the mental nerve, which arises from the mandibular division (V3), the probe is positioned 1 cm superior to the inferior border of the mandible and approximately 3 cm lateral to the midline (Figure 1A,D). The mental foramen is identified as a gap in the bony cortex and typically contains the mental nerve accompanied by vessels.

For maxillary nerve blockade, the patient is placed in the lateral decubitus position with the affected side facing up. A curvilinear transducer is placed parallel to the zygomatic arch to visualize the lateral pterygoid muscle and the lateral pterygoid plate of the sphenoid bone (Figure 2A,B). Doppler imaging facilitates identification of the sphenopalatine artery. The needle is inserted using an out-of-plane approach to reach the pterygopalatine fossa. To access the mandibular nerve, the patient remains in the same lateral position. The needle is directed from anterior to posterior, targeting the space just posterior to the lateral pterygoid plate and between the medial and lateral pterygoid muscles (Figure 2A,B).

## 4. Postherpetic Neuralgia

Postherpetic neuralgia is a chronic complication of herpes zoster. It occurs in up to one-third of patients, with the incidence increasing markedly with age [24]. Those with prodromal pain, severe pain, or extensive skin rash are also more likely to be affected [25]. Early intervention with sympathetic and somatic neural blocks during the acute phase may lower the risk of postherpetic neuralgia development [26]. Other options under exploration include subcutaneous injections and paravertebral blocks.

Several studies have found that epidural corticosteroid injections combined with local anesthetics significantly reduced visual analog scale (VAS) scores in both acute and chronic herpes zoster [27,28,29]. Compared to fluoroscopy-guided epidural nerve blocks, erector spinae plane (Figure 3A,B) and paravertebral blocks are technically easier to achieve under ultrasound guidance and do not require an operating room or fluoroscopic equipment. A recent systematic review and meta-analysis demonstrated that both nerve block techniques significantly reduced pain severity and decreased the incidence of postherpetic neuralgia at three and six months after the procedure [30].

Paravertebral injections appear to carry a higher risk of complications, with reported adverse events including dizziness, drowsiness, and localized pain at the injection site. Deng et al. [31] elaborated two ultrasound-guided paravertebral block techniques for treating acute thoracic postherpetic pain using 2 mL of injectate (2% lidocaine + triamcinolone + normal saline). Both techniques yielded comparable outcomes in improving quality of life and reducing postherpetic neuralgia incidence at six-month follow-up. For patients undergoing the parasagittal oblique approach, a linear transducer was positioned sagittally, 2–3 cm lateral to the thoracic paramedian line (Figure 4A,B). The paravertebral muscles and adjacent hyperechoic transverse processes were revealed. The acoustic window—bounded by the costotransverse ligament, superior costotransverse ligament, thoracic paravertebral space, and parietal pleura—enabled safe needle advancement in-plane from caudal to cranial. For the transverse short-axis approach, a probe was placed horizontally over the lateral aspect of the thoracic spinous process (Figure 4A,C). The paravertebral muscles and hyperechoic transverse process were visualized, and by slight cranial or caudal probe adjustments, the paravertebral space could be identified. Color Doppler was used to avoid vascular injury. A 22-gauge needle was inserted in-plane from lateral to medial.

A case report illustrated selective cervical nerve root block at the C5 (Figure 5A,B) and C6 (Figure 5C,D) levels using a mixture of 0.4 mL 50% dextrose and 3.6 mL 1% lidocaine for upper arm postherpetic pain [32]. The patient was positioned supine with the head turning away from the affected side. The transducer was placed horizontally, and the needle was advanced in-plane from posterior to reach the posterior tubercle, surrounding the nerve root with the injectate. Following the procedure, the patient’s VAS score decreased from 8 cm to 2 cm on a 10 cm scale. Moon et al. [33] injected 10 mL 0.1% bupivacaine with 50 units of botulinum toxin type A around the brachial plexus to treat upper limb postherpetic pain. Effective pain control was maintained throughout the five-month follow-up period. The effect of an ultrasound-guided satellite ganglion block for facial postherpetic pain has also been evaluated, with results indicating possible benefits (Figure 6A,B) [34,35]. The needle was advanced in-plane along a lateral-to-medial trajectory delivering the injectate anterior to the longus colli muscle and posterior to the internal carotid artery and jugular vein. Doppler assessment was crucial to avoid vascular injury given the close proximity of large vessels to the target site.

Some investigations have employed ultrasound- or fluoroscopy-guided pulsed radiofrequency (PRF) of the dorsal root ganglion for postherpetic neuralgia, demonstrating superior analgesic outcomes compared to pharmacologic therapy alone [36,37]. Wang et al. [38] found no significant differences in pain scores, sleep quality, or oral medication use between the two methods for thoracic lesions over a 24-week follow-up; however, the ultrasound-guided approach was associated with reduced radiation exposure and shorter procedure time. For real-time ultrasound guidance, the transducer is oriented in the parasagittal plane with the patient lying prone. Once the intertransverse ligament between adjacent transverse processes is visualized, the needle is advanced under in-plane guidance toward the region of the posterior primary ramus. Pi et al. [39] allocated 128 patients to receive either pharmacologic therapy alone or in combination with ultrasound-guided posterior ramus PRF via paravertebral puncture. The PRF group achieved significantly greater pain reduction, with a mean decrease in VAS score of 6.2 cm compared with 4.5 cm in the control group on a 10 cm scale, along with reduced morphine consumption over a two-month period.

PRF could also be applied for facial postherpetic neuralgia. In the case report by Lim et al. [40], ultrasound-guided PRF was performed to the infra-orbital nerve in a patient with intractable postherpetic neuralgia. Following treatment, the patient’s VAS score declined drastically from 9 to 10 to 1–2, with sustained pain relief for one year. Li et al. [41] studied 91 patients with craniofacial postherpetic neuralgia whereby ultrasound-guided PFA (for supraorbital, infraorbital, mental, and greater occipital nerves) significantly reduced mean VAS from 6.7 cm to 2.2 cm on a 10 cm scale, and enhanced quality of life and sleep quality. Hypoesthesia was the most common complication reported in 82.8% of the patients. A comparative study on ultrasound-guided nerve block with and without PRF for V1 postherpetic neuralgia showed superior outcomes with PRF [42]. Among 62 patients, response rates (defined as >25% pain reduction) were 25% for nerve block alone and 32% for the PRF group. Statistically lower VAS scores were observed in the PFR group at one-, three-, and six-month post-treatment.

In a case report, a man with refractory left facial postherpetic neuralgia received supraorbital peripheral nerve stimulation via an electrode implanted under combined ultrasound and fluoroscopy guidance [43]. At nine-month follow-up, the patient’s average weekly VAS score had decreased from 8 to 1.

## 5. Post-Amputation/Mastectomy Pain

Pain among amputees is both prevalent and multifactorial. Contributing factors include poor circulation, skin breakdown, surgical wounds, and poorly fitted prosthetics. Neuropathic pain is also a major concern in this population and may manifest as neuroma-related pain or phantom limb pain.

Neuromas are formed due to abnormal nerve regeneration after a peripheral nerve is transected or by repetitive irritation. Clinically, they present as small, tender, palpable masses associated with numbness, tingling, or localized sharp pain. Ultrasound enables their identification, and by rotating the transducer, the proximal trajectory of the parent nerve can often be traced, aiding in accurate localization. Ultrasound is also helpful in arranging perineural injections. In one study, ultrasound-guided corticosteroid injections were given to neuromas in 14 lower limb amputees, and 50% of them had successful results (>50% reduction in pain) [44]. In a below-elbow amputee, an ultrasound-guided injection of 1 mL triamcinolone (10 mg/mL) and 1 mL 1% lidocaine around median and ulnar nerve neuromas led to symptom relief and shrinkage in the neuroma size [45]. Ultrasound-guided PRF could be another safe and effective way to treat stump neuromas and promote prosthetics tolerance [46,47]. In a study by Pu et al. [46], 82.4% of 17 patients experienced favorable outcomes following ultrasound-guided neuroma PRF.

Phantom limb pain refers to painful sensations perceived in the missing body part. Its pathophysiology is believed to arise from maladaptive central and peripheral neural plasticity, including cortical reorganization and spinal sensitization [48]. Oral medication remains the mainstay management approach, but there is growing interest in injection-based therapies. A prior study showed that a single axillary brachial plexus block using 40 mL 1% mepivacaine significantly reduced pain and was associated with changes in cortical remapping in patients with upper limb phantom pain [49]. However, ultrasound-guided percutaneous cryoneurolysis did not produce substantial amelioration in phantom limb pain in the randomized controlled trial by IIfeld et al. [50], but the level of amputation may have influenced the outcome. Additionally, ultrasound may aid in preemptive analgesic procedures like regional nerve blocks in the hopes of decreasing the incidence of phantom limb pain, but more research is needed to establish definite efficacy.

In a randomized, double-blind, placebo-controlled trial, peripheral nerve stimulation was compared with sham stimulation in 28 lower-extremity amputees with chronic neuropathic pain [51]. Percutaneous leads were placed under ultrasound guidance targeting the femoral and sciatic nerves, with continuous stimulation administered for eight months. A significantly larger proportion of participants receiving active therapy reported ≥50% pain relief compared to those in the placebo group during the stimulation period. Notably, 80% of treated participants continued to experience ≥50% pain relief—averaging a 76% reduction in pain—10 months after discontinuing stimulation.

Likewise, patients undergoing mastectomy may experience neuropathic pain due to intercostobrachial nerve injury, intercostal cutaneous branch neuromas, or phantom breast pain [52]. The serratus plane block (Figure 7A,B) has been proposed as a component of pain management for persistent pain following mastectomy [53,54]. Ultrasound can ensure precise needle placement and minimize complications such as pneumothorax. Dayem et al. [55] administered ultrasound-guided stellate ganglion block three times at a one-week interval in patients with postmastectomy pain persisting for more than six months. The patient lies supine with the head turned contralaterally and a high-frequency linear probe is placed transversely at the C6 level to visualize the carotid artery, longus colli muscle, and the transverse process with its prominent anterior tubercle (Chassaignac’s tubercle). Using an in-plane, lateral-to-medial approach, a 22-gauge needle is advanced toward the Chassaignac tubercle. After contacting the tubercle, the needle is withdrawn 1–2 mm to position the tip just outside the longus colli muscle. Following negative aspiration for blood, the drug particles were deposited within the prevertebral fascia. The intervention resulted in significant pain relief, decreased need for pain medication, and increased shoulder range of motion.

## 6. Painful Polyneuropathy and Peripheral Nerve Injury Pain

There are many causes of painful neuropathy, e.g., diabetes, infections, alcoholism, renal disease, connective tissue disorders, inherited neuropathies, toxic exposures, nerve injuries, etc. As of now, ultrasound-guided intervention is infrequently reported as an adjuvant/alternative treatment option in these scenarios.

Diabetes is the leading cause of peripheral neuropathy, with most patients developing this complication over time. The most prevalent form is distal symmetrical polyneuropathy, characterized by progressive sensory loss starting from the distal lower extremities, although other patterns may also occur [56]. Management focuses on glycemic control and symptomatic pain relief. Lumbar plexus block (Figure 8A,B) using botulinum toxin type A [33] yielded significant relief of severe diabetic leg pain for four months. For patients with refractory painful distal symmetrical polyneuropathy, ultrasound-guided PRF of the stellate ganglion has been reported to show clinical benefits. In one study, the effective pain relief rates (≥50% reduction in pain) were 67.86%, 42.86%, 21.43%, and 17.86% at 1, 4, 12, and 24 weeks post-treatment, respectively, [57]. Hu et al. [58] employed an innovative ultrasound-guided hydrodissection approach. The procedure was carried out at multiple sites, including the bifurcation of the tibial and fibular nerves (Figure 9A,B), the peroneal nerve at the fibular head (Figure 9A,C), and the tibial nerve at the medial malleolus (Figure 9A,D). They proposed that this technique could release nerve adhesions and boost venous and lymphatic outflow, thereby alleviating paresthesia and pain. Pain improved in all five patients.

Ultrasound plays a valuable role in both diagnosing the origin and guiding the treatment of peripheral neuropathic pain by enabling precise assessment of the injury’s location, extent, and severity. Wang et al. [59] reported a case of successful treatment for bilateral sciatic nerve injury following prolonged surgery by delivering a perineural injection at the level of the superior gemellus muscle (Figure 10A,B). Both patients were satisfied with their improved sleep and quality of life. Kim et al. [60] applied peripheral nerve stimulation in two patients with severe neuropathic pain following brachial plexus injury. After ultrasound-guided electrode placement, both patients experienced over 50% sustained pain relief over one year. Narouze et al. [61] successfully treated a patient with iatrogenic femoral nerve injury and neuropathic pain with peripheral nerve stimulation. After ultrasound-guided and fluoroscopy-confirmed percutaneous lead placement for stimulation, the patient remained pain-free for 15 months. A detailed description of the neuroanatomy relevant to knee pain was published by Lin et al. [62], offering guidance for peripheral nerve stimulation and other related procedures. Chen et al. [63] treated a case of traumatic brachial plexus injury with a C6 selective nerve root block using a mixture of 1 mL dexamethasone (5 mg/mL) and 5 mL methylcobalamin. The VAS score improved from 8 to 3 on a 10 cm scale.

## 7. Painful Radiculopathy

Radiculopathy presents as pain, tingling, clumsiness, and even weakness in the distribution of the affected nerve root. Conservative management includes patient education, manual therapy, exercise, medications such as paracetamol, non-steroidal anti-inflammatory drugs, short-term opioids, and cervical traction [64]. The next step is often epidural injections, particularly for patients with severe pain or those who are reluctant to undergo surgery. Anesthetic and corticosteroid injections block inflammatory responses and may decrease the volume of disc herniation. Ultrasound-guided cervical transforaminal epidural injections appeared to be safe and effective in decreasing pain and disability for cervical radiculopathy [65]. In patients with lumbar radiculopathy, two meta-analyses have indicated that such interventions can provide pain relief and functional improvement for three months [66,67]. The efficacy of ultrasound-guided epidural injection for radiculopathy might be parallel to that achieved with conventional fluoroscopy-guided techniques [68]. The ultrasound group may have the additional advantage of less inadvertent vascular puncture and shorter procedure time.

Pararadicular spinal blocks target a single nerve root at a time—hence the term “selective nerve root block”—and are less invasive than epidural injections. Some studies suggest that it may offer similar efficacy to transforaminal epidural injection in relieving pain and improving function in patients with cervical radiculopathy [69,70]. Lee et al. [71] treated 49 patients with chronic cervical radiculopathy irresponsive to epidural steroid injections; ultrasound-guided selective nerve root PRF provided sustained pain relief for six months in 63.3% of the cases.

Wang et al. [72] retrospectively compared the effects of ultrasound- and fluoroscopy-guided lumbar selective nerve root blocks in patients with lumbar nerve root pain. Both groups showed matching ability to lessen pain, with mean decreases in VAS scores of 4.5 cm and 4.3 cm on a 10 cm scale, respectively, for six months. But the ultrasound group needed significantly shorter operation time and fewer needle angle adjustments. Two ultrasound-guided lumbar selective nerve root block techniques for patients with lumbar radiculopathy were compared by Kim et al. [73]. In the median plane, the transducer was placed longitudinally to delineate the transition from the fifth lumbar to the first sacral spinous process. It was then moved laterally to visualize the “trident sign” formed by two adjacent transverse processes, with the intertransverse ligament appearing as a hyperechoic band connecting them [74]. The pararadicular compartment lay between this ligament and the psoas muscle. In the paramedian sagittal approach, the transducer was positioned perpendicular to the skin whereas in the paramedian sagittal oblique approach, it was angled about 20–25° medially after identifying the intertransverse ligament. The needle was advanced in-plane in both approaches. Although both techniques significantly reduced pain at four-weeks post-injection, the oblique approach exhibited higher accuracy and greater pain relief. The effectiveness of ultrasound- (Figure 11A,B) vs. fluoroscopy-guided caudal epidural injections for lumbar radicular pain has been evaluated in several studies and implied comparable outcomes [75,76,77]. In the randomized controlled trial by Elashmawy et al. [77], 121 patients received a 20 mL injectate consisting of 18 mL 0.5% lidocaine and 2 mL triamcinolone acetonide (40 mg/1 mL) by caudal route either by ultrasound or fluoroscopy guidance. The patients were placed in the prone position, and after meticulous aseptic preparation, a linear ultrasound probe was positioned transversely at the midline to identify the two sacral cornua and the hyperechoic sacrococcygeal ligament overlying the sacral hiatus. The probe was then rotated 90° to obtain a longitudinal view, and the needle was advanced in-plane through the sacrococcygeal ligament to access the epidural space. VAS scores were significantly reduced in both groups at one-month and three-month post-procedure, with no intergroup difference detected at the two timepoints.

## 8. Discussion

This narrative review compiles ultrasound-guided interventions for neuropathic pain and presents high-resolution sonographic figures to highlight key procedural techniques (Table 1). Research on this topic has expanded in recent years, particularly regarding postherpetic neuralgia and painful radiculopathy. The most frequently employed interventions include ultrasound-guided nerve blocks, PRF, and epidural injections for the management of painful radiculopathy. Early reports suggest that these approaches may offer meaningful improvements in pain control, functional outcomes, and quality of life, though the current data remains preliminary.

The majority of current guidelines for neuropathic pain place emphasis on stepwise pharmacological therapy [78,79,80,81]. As for interventional therapies, the most consistent evidence to date pertains to herpes-related pain and radiculopathy, which is consistent with our synthesis. According to the recommendations of the International Association for the Study of Pain Neuropathic Pain Special Interest Group, moderate-quality evidence supports the use of epidural or paravertebral nerve blocks for acute herpes zoster pain and epidural injections for lumbar radiculopathy [82]. Nonetheless, a recent network meta-analysis reported minimal improvement by epidural injections or PRF for chronic radiculopathy [83]. Actually, evidence is relatively stronger for radicular pain secondary to disc herniation than for radiculopathy associated with spinal stenosis [84]. These findings highlight the necessity of accurate diagnosis, careful patient selection, procedural expertise, and comprehensive informed consent regarding the potential benefits and limitations of injection-based therapies. Evidence for interventional treatments in trigeminal neuralgia, postherpetic neuralgia, and painful diabetic peripheral neuropathy is of low quality and insufficient to justify recommendations [82]. Another recommendation published in 2020 stated that the evidence was inconclusive concerning the efficacy of nerve blocks and intrathecal injections for neuropathic pain [81]. In addition, moderate evidence supports only a weak recommendation for transcutaneous electrical nerve stimulation (TENS) in peripheral neuropathic pain and for PRF in thoracic postherpetic neuralgia [81]. In their proposed treatment algorithm for peripheral neuropathic pain, TENS was recommended as a first-line therapy, while subcutaneous botulinum toxin A was designated as a second-line option. Notably, the treatment ladder omitted both steroid and local anesthetic injections, and the guidelines did not indicate a preference between ultrasound and fluoroscopic guidance.

Ultrasound guidance may offer certain several benefits over fluoroscopy, though both techniques have distinct roles. First, fluoroscopy-guided procedures require repeated use of X-ray imaging and contrast dye injections to confirm instrument placement. The mean radiation exposure was calculated as 5.6 mGy for cervical transforaminal injections and 10.4 mGy for lumbar transforaminal injections [85]. The International Commission on Radiological Protection sets the annual equivalent dose limit for the skin at 500 mSv and the effective dose limit for occupational exposure at 50 mSv. The cumulative lifetime risk of radiation-induced fatal cancer varies with total dose and age at exposure; for a 40-year-old adult, the estimated threshold for a measurable increased risk is approximately 700 mSv [86]. These thresholds underscore the importance of radiation precautions for clinicians who routinely perform fluoroscopy-guided procedures. Ultrasound guidance eliminates radiation exposure and is particularly advantageous for vulnerable populations, such as pregnant women [87]. Second, grave consequences, including arterial dissection and death, have been reported following fluoroscopy-guided cervical nerve root blocks [88]. Inadvertent intravascular injection of corticosteroids or anesthetics may also lead to seizures, allergic reactions, transient paresis, or infarctions [85,89]. Ultrasound boosts the capability to clearly display blood vessels, nerves, and other soft tissue structures, thereby reducing the risk of vascular puncture and unintended drug delivery into the bloodstream. With real time ultrasound imaging, experienced physicians can promptly adjust the needle trajectory and angle, minimizing tissue trauma and enhancing patient comfort. It should be emphasized that there is a lack of high-certainty evidence confirming equivalent efficacy of ultrasound versus fluoroscopy guidance, especially for spinal injections [90,91,92]. Moreover, the accuracy of ultrasound-guided procedures is highly machine- and operator-dependent; it may take years of training before novice practitioners feel confident in these techniques. For greater precision and safety, a dual-guidance approach—utilizing both ultrasound and fluoroscopy—may often be advisable [51,93,94].

Neuropathic pain is a challenging clinical conundrum, due in part to its complicated pathophysiology and in part to the frequent overlap with non-neuropathic pain syndromes. Translational research is further hindered by the limited applicability of animal models to effective human treatment. Only 30–40% of patients achieved adequate response compared with placebo [95]. Consequently, emerging strategies include drug repositioning, neurostimulation techniques, and individualized pain management [11]. A successful example of drug repositioning is botulinum toxin type A—originally used for treating spasticity and dystonia through blocking acetylcholine release at the neuromuscular junction [96]. Pain relief begins before any measurable reduction in muscle activity and is more pronounced than those attributable to muscle relaxation alone [97]. This antinociceptive effect, independent of muscle tone, is hypothesized to involve central mechanisms mediated through retrograde axonal transport [98]. Methylcobalamin (vitamin B12) is another promising injectable agent for neuropathic pain and peripheral nerve injuries. It is thought to exert analgesic effects by promoting nerve regeneration, upregulating brain-derived neurotrophic factor, and inhibiting cyclooxygenase enzymes [99]. Although traditionally administered orally, its efficacy via subcutaneous injection has also been documented in alleviating symptoms of diabetic peripheral neuropathy, postherpetic neuralgia, and chemotherapy-induced peripheral neuropathy [99,100]. The most effective injectates for each subtype of neuropathic pain warrant more thorough investigation.

The precise mechanism by which PRF alleviates pain is still incompletely understood. Typical regimens use a frequency of 2 to 5 Hz and a pulse width of 5 to 20 ms, delivered for 240 to 360 s, with the electrode tip temperature kept below 42 °C. PRF could enhance noradrenergic and serotonergic descending pain inhibitory pathways, down-regulate excitatory C-fiber activity, and suppress pro-inflammatory cytokines like tumor necrosis factor-α and interleukin-6 [101,102]. By modulating pain signals and decreasing nociceptive free nerve endings, PRF may stop the progression of chronic neuropathic pain. It produces longer-lasting analgesic effects than nerve blocks, often extending several months [103,104]. On the other hand, nerve blocks are performed with local anesthetics, with or without corticosteroids. When local anesthetics are used alone, the duration of pain relief depends on the pharmacokinetics of the chosen agent and adjuvants, but it rarely exceeds eight hours. Even with the addition of corticosteroids, the duration of analgesia is still shorter than that achieved with PRF [104,105].

This review is not free from limitations. First, the included studies showed substantial heterogeneity in patient selection, procedural techniques, and injectate formulations. Second, follow-up was generally limited to 3–6 months, and data on long-term outcomes are lacking. Third, many studies were retrospective in design or consisted of case reports and small series. Even among randomized controlled trials, the absence of double-blinding often introduced a huge risk of bias. Furthermore, few studies documented the use of optimal pharmacologic therapy in enrolled participants, and no head-to-head comparisons were conducted. Lastly, due to the inherent nature of narrative reviews, formal quality assessments of included studies were not performed, rendering the synthesis susceptible to subjective interpretation. Future progress will depend on the development of standardized protocols, informed by multicenter studies, expert recommendations, and robust trial design, to ensure stronger evidence and broader clinical uptake.

## 9. Conclusions

Ultrasound-guided, minimally invasive interventions have demonstrated potential in alleviating various forms of neuropathic pain. With advancing technology and growing knowledge of the field, these techniques may play an increasingly important role in pain management. However, current evidence remains insufficient to establish standardized algorithms, and further research is needed to validate their efficacy and compare them with conventional treatments.

## Figures and Tables

**Figure 1 life-15-01404-f001:**
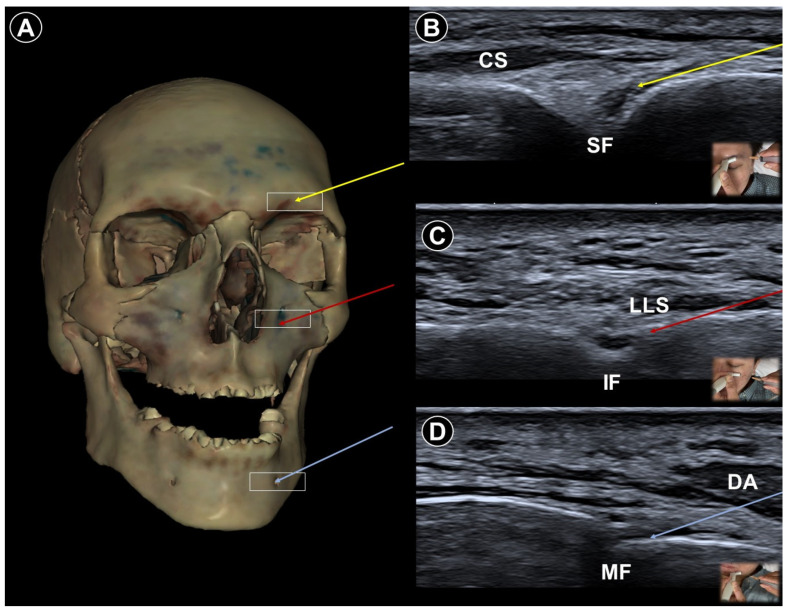
Cadaveric bony model reconstruction illustrating the supraorbital foramen (SF), infraorbital foramen (IF), and mental foramen (MF) (**A**), and ultrasound-guided injections targeting the supraorbital nerve (V1; needle trajectory indicated by the yellow arrow) (**B**), infraorbital nerve (V2; needle trajectory indicated by the red arrow) (**C**), and mental nerve (V3; needle trajectory indicated by the blue arrow) (**D**). CS, corrugator supercilii; LLS, levator labii superioris; DA, depressor anguli oris. The white square indicates the transducer footprint. Cadaveric images adapted from cadaveric images provided by the Visible Human Project^®^ of the National Library of Medicine. Excerpts featured in the VH Dissector are used with permission from Touch of Life Technologies Inc. (Aurora, CO, USA).

**Figure 2 life-15-01404-f002:**
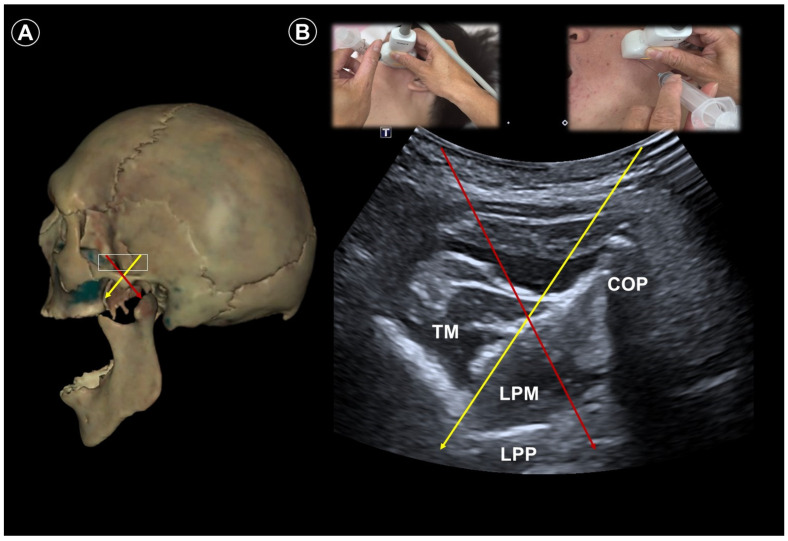
Cadaveric bony model reconstruction (**A**), and ultrasound images (**B**), illustrating ultrasound-guided injections targeting the maxillary nerve (V2; needle trajectory indicated by the yellow arrow) and mandibular nerve (V3; needle trajectory indicated by the red arrow). TM, temporalis muscle; LPM, lateral pterygoid muscle; COP, condylar process; LPP, lateral pterygoid plate. The white square indicates the transducer footprint. Cadaveric images adapted from cadaveric images provided by the Visible Human Project^®^ of the National Library of Medicine. Excerpts featured in the VH Dissector are used with permission from Touch of Life Technologies Inc.

**Figure 3 life-15-01404-f003:**
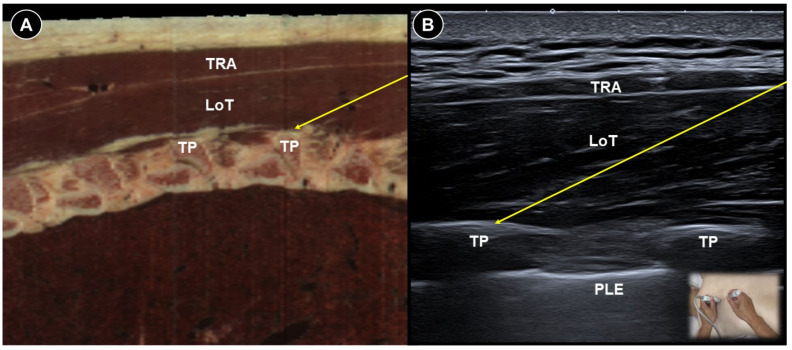
Cadaveric sagittal cross-sectional image (**A**), and corresponding ultrasound image (**B**), illustrating an ultrasound-guided erector spinae plane block, with the needle trajectory indicated by the yellow arrow. TRA, trapezius; LoT, longissimus thoracis; TP, transverse process; PLE, pleura. Cadaveric images adapted from cadaveric images provided by the Visible Human Project^®^ of the National Library of Medicine. Excerpts featured in the VH Dissector are used with permission from Touch of Life Technologies Inc.

**Figure 4 life-15-01404-f004:**
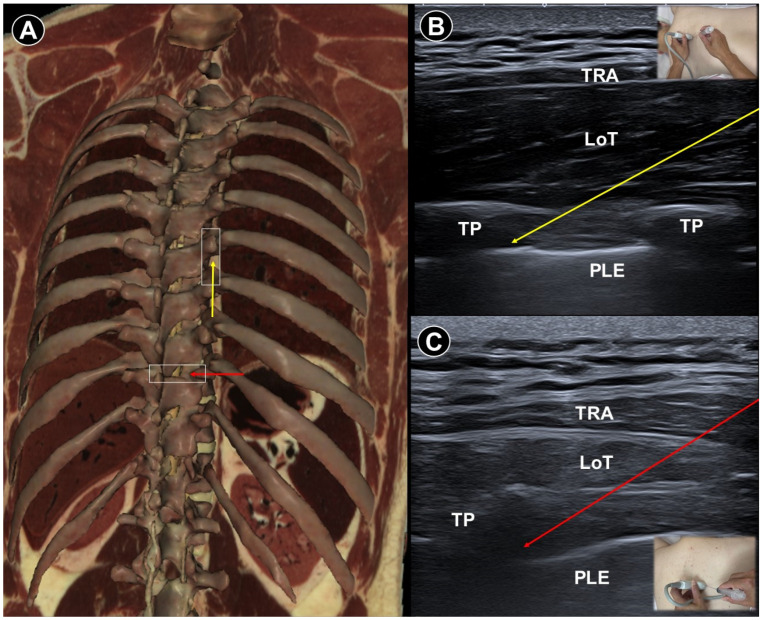
Cadaveric bony model reconstruction (**A**), and ultrasound images illustrating two approaches for paravertebral block: the parasagittal oblique approach (**B**) (needle trajectory indicated by the yellow arrow), and the transverse short-axis approach (**C**) (needle trajectory indicated by the red arrow). TRA, trapezius; LoT, longissimus thoracis; TP, transverse process; PLE, pleura. The white square indicates the transducer footprint. Cadaveric images adapted from cadaveric images provided by the Visible Human Project^®^ of the National Library of Medicine. Excerpts featured in the VH Dissector are used with permission from Touch of Life Technologies Inc.

**Figure 5 life-15-01404-f005:**
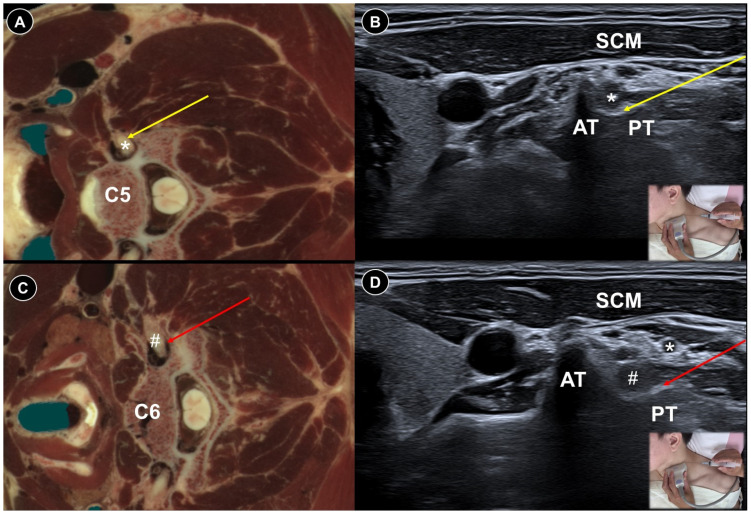
Cadaveric axial-sectional image (**A**), and corresponding ultrasound image (**B**), illustrating ultrasound-guided injection of the C5 nerve root (asterisk), with the needle trajectory indicated by the yellow arrow. Cadaveric axial-sectional image (**C**), and corresponding ultrasound image (**D**), illustrating ultrasound-guided injection of the C6 nerve root (hashtag), with the needle trajectory indicated by the red arrow. AT, anterior tubercle; PT, posterior tubercle; SCM, sternocleidomastoid.

**Figure 6 life-15-01404-f006:**
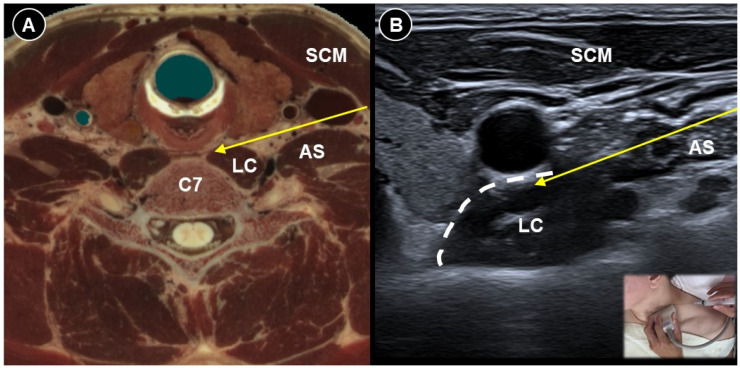
Cadaveric axial-sectional image (**A**), and corresponding ultrasound image (**B**), illustrating ultrasound-guided stellate ganglion block. The needle trajectory (yellow arrow) targets the region deep to the prevertebral fascia (dashed white line) overlying the longus colli (LC) muscle. AS, anterior scalene; SCM, sternocleidomastoid.

**Figure 7 life-15-01404-f007:**
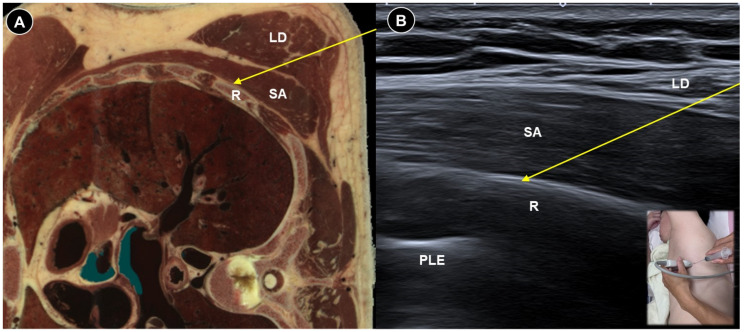
Cadaveric axial-sectional image (**A**), and corresponding ultrasound image (**B**), illustrating an ultrasound-guided serratus anterior (SA) plane block, with the needle trajectory indicated by the yellow arrow. LD, latissimus dorsi; R, rib; PLE, pleura. Cadaveric images adapted from cadaveric images provided by the Visible Human Project^®^ of the National Library of Medicine. Excerpts featured in the VH Dissector are used with permission from Touch of Life Technologies Inc.

**Figure 8 life-15-01404-f008:**
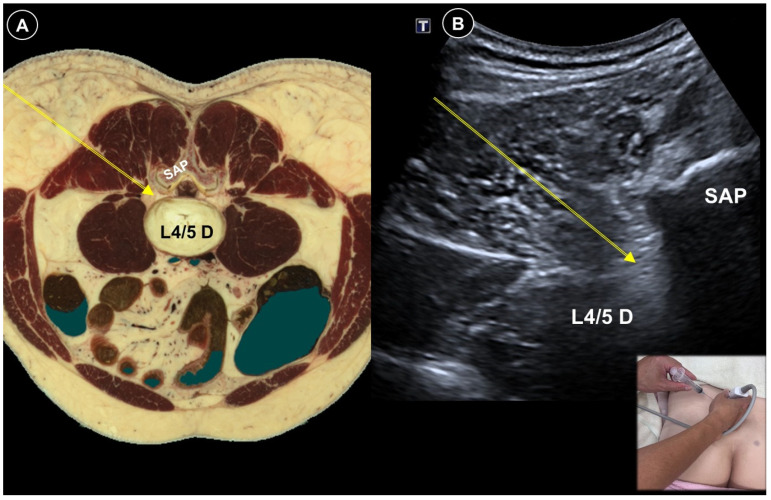
Cadaveric axial-sectional image (**A**), and corresponding ultrasound image (**B**), illustrating an ultrasound-guided lumbar plexus block, with the needle trajectory indicated by the yellow arrow. SAP, superior articular process; L4/5 D, intervertebral disc between L4 and L5. Cadaveric images adapted from cadaveric images provided by the Visible Human Project^®^ of the National Library of Medicine. Excerpts featured in the VH Dissector are used with permission from Touch of Life Technologies Inc.

**Figure 9 life-15-01404-f009:**
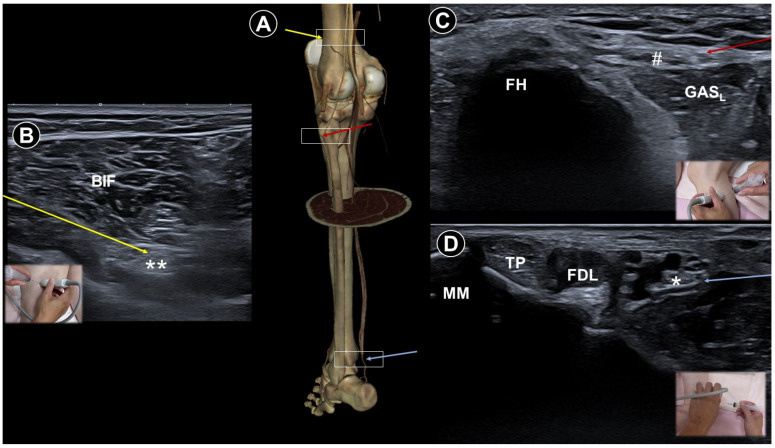
Cadaveric bony model reconstruction illustrating the sciatic nerve of the lower extremity (**A**), and ultrasound-guided injections targeting the sciatic nerve (double asterisks; needle trajectory indicated by the yellow arrow) near its distal bifurcation (**B**), the common peroneal nerve (hashtag; needle trajectory indicated by the red arrow) near the fibular head (FH) (**C**), and the tibial nerve (single asterisk; needle trajectory indicated by the blue arrow) near the medial malleolus (MM) (**D**). BIF, biceps femoris; GAS_L_, lateral gastrocnemius; TP, tibialis posterior; FDL, flexor digitorum longus. Cadaveric images adapted from cadaveric images provided by the Visible Human Project^®^ of the National Library of Medicine. Excerpts featured in the VH Dissector are used with permission from Touch of Life Technologies Inc.

**Figure 10 life-15-01404-f010:**
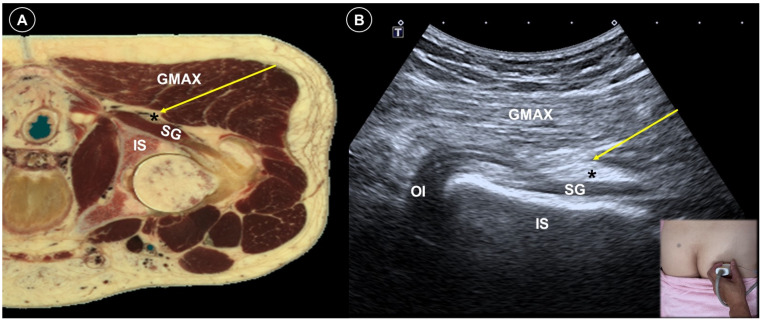
Cadaveric axial-sectional image (**A**), and corresponding ultrasound image (**B**), illustrating an ultrasound-guided sciatic nerve (asterisk) block, with the needle trajectory indicated by the yellow arrow. GMAX, gluteus maximus; SG, superior gemellus; OI, obturator internus; IS, ischium. Cadaveric images adapted from cadaveric images provided by the Visible Human Project^®^ of the National Library of Medicine. Excerpts featured in the VH Dissector are used with permission from Touch of Life Technologies Inc.

**Figure 11 life-15-01404-f011:**
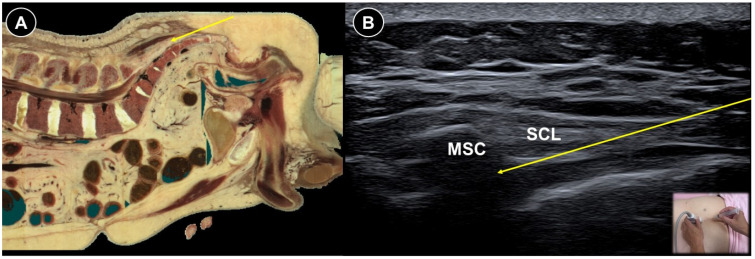
Cadaveric sagittal section (**A**), and corresponding ultrasound image (**B**), illustrating a caudal block, with the needle trajectory indicated by the yellow arrow. MSC, median sacral crest; SCL, sacrococcygeal ligament. Cadaveric images adapted from cadaveric images provided by the Visible Human Project^®^ of the National Library of Medicine. Excerpts featured in the VH Dissector are used with permission from Touch of Life Technologies Inc.

**Table 1 life-15-01404-t001:** Summary of review findings. N, nerve; VZV, varicella-zoster virus; SNRB, selective nerve root block; SGB, satellite ganglion block; TPVB, thoracic paravertebral block; ESPB, erector spinae plane block; PRF, pulsed radiofrequency; RCT, randomized controlled trial; PNS, peripheral nerve stimulation; DPN, diabetic polyneuropathy; ESI, epidural steroid injection.

	Pain Characteristic	Cause	Patient Positioning	Ultrasound Probe and Technique	Effectiveness of Ultrasound-Guided Procedures	Additional Notes
**Trigeminal neuralgia**	Paroxysmal facial pain in one or more divisions of the trigeminal nerve	Neurovascular compression at root entry zone	Head neutral	Linear probe, in-plane V1: supraorbital notch V2: infraorbital foramen V3: mental foramen	Sustained pain relief by nerve block (case study)	Use Doppler imaging to avoid vascular puncture
			Lateral, with affected side up	Curvilinear probe, out-of-plane Maxillary N.: pterygopalatine fossa Mandibular N.: between pterygoid muscles		
**Acute herpes pain**	Burning, stabbing, or itching pain in the affected dermatomal distribution	Reactivation of VZV leading to viral nerve damage and inflammation	Cervical: supine, head turned to contralateral side Thoracic: prone	Cervical: SNRB brachial plexus block SGB Thoracic: TPVB ESPB	Reduce pain and lower postherpetic neuralgia incidence by TPVB/ESPB (meta-analysis), PRF/SNRB/SGB (RCT)	Caution near large vessels (cervical) and lungs (thoracic) Prefer non-particulate steroids
**Post-herpetic neuralgia**	Persistent pain after rash resolution	Central and peripheral sensitization			Effective pain control by PRF (RCT), TPVB/SNRB/brachial plexus block/SGB/PNS (case study)	
**Post-amputation pain**	Stump pain and phantom limb pain	Multifactorial, including neuromas, nerve damage, sensitization	Variable depending on the target nerve Neuroma identified as a hypoechoic, swollen mass adjacent to the parent nerve		Significant pain reduction by PNS (RCT), nerve block/PRF (case study)	Avoid direct injection into the nerve
**Painful polyneuropathy**	Numbness, tingling sensations, dysesthesia, weakness, balance impairment	Metabolic, infections, connective tissue disorders, hereditary, toxins, nerve injuries, etc.	Variable depending on the target nerve Commonly distal sensory nerves in lower extremities		Improvement of painful DPN by SGB (prospective cohort), lumbar plexus block/hydrodissection (case study) Peripheral nerve injury pain decreased by nerve blocks/PNS (case study)	Ultrasound facilitates targeting small distal nerves
**Painful radiculopathy**	Pain, numbness, clumsiness and even weakness in the distribution of the affected nerve root	Mechanical compression and inflammatory irritation of the nerve root, often due to disc herniation or spondylosis	Cervical: supine, head turned to contralateral side Lumbar: prone	Cervical: linear, in-plane Transforaminal ESI SNRB Lumbar: curvilinear probe for ESI/SNRB, linear probe for caudal block, in-plane Transforaminal ESI SNRB Caudal block	Improvement of pain by ESI (meta-analysis), SNRB/caudal block (RCT)	Caution in cervical region with large vessels Prefer non-particulate steroids

## Data Availability

No new data were created or analyzed in this study. Data sharing is not applicable to this article.

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
