# Peer review of "Ultrasound-Guided Interventions for Neuropathic Pain: A Narrative Pictorial Review"

_life, 2025, doi:10.3390/life15091404_

Round 1
Reviewer 1 Report
Comments and Suggestions for Authors
This chapter is very enthusiastically pro-ultrasound guided nerve blocks without clear evidence of actual benefit. A careful look at some of the references reveals exaggeration of results reflecting the same enthusiasm.
In the abstract the authors comment on the frailty of the literature, but in the main body of the text the tone changes and many of the results are described as encouraging (which is not in fact true).
Enthusiasm over evidence.
It may be that ultrasound-guidance does have a particular benefit beyond radiation safety aspects (exposure is low as it is with use of fluoroscopy) but this chapter does not prove that.
As an experienced clinician, I also disagree strongly with the comments that it is more accurate than fluoroscopy. We are talking apples and oranges here, with different technologies for different purposes.
Some of the sections mix pathologies (eg Trigeminal Neuralgia section) and this is unhelpful.
It is in complete contrast to the recent articles and editorial in BMJ Open which suggest that injection therapies per se do not have therapeutic value. This was not addressed and is of major concern. The danger is the encouragement of others to 'have a go' when evidence is poor or even lacking
Author Response
Reviewer 1
Comment:
In the abstract the authors comment on the frailty of the literature, but in the main body of the text the tone changes and many of the results are described as encouraging (which is not in fact true). Enthusiasm over evidence.
Response:
We agree with the reviewer’s concern. We have emphasized the paucity of high-quality evidence and the need for further research in the abstract (Notably, much of the existing literature comprises small-scale or observational studies and larger randomized controlled trials are therefore essential to confirm efficacy, define optimal treatment parameters, and inform clinical guidelines for broader adoption.), discussion (Early reports suggest that these approaches may offer meaningful improvements in pain control, functional outcomes, and quality of life, though the current data remains pre-liminary.), and conclusion(However, current evidence remains insufficient to establish standardized algorithms, and further research is needed to validate their efficacy and compare them with conventional treatments.) Additionally, in the limitations section, we wrote: “Lastly, due to the inherent nature of narrative reviews, formal quality assessments of included studies were not performed, rendering the synthesis susceptible to subjective interpretation.”
The positive findings presented in this review are derived directly from previous studies have been described with appropriate caution to avoid overstating their significance.
We have also adjusted the wording in the lines listed below to avoid exaggeration.
Line 213: The effect of ultrasound-guided satellite ganglion block for facial postherpetic pain has also been evaluated, with results indicating possible benefit.
Line 276: Ultrasound-guided PRF could be another safe and effective way to treat stump neuromas and promote prosthetics tolerance
Line 336: For patients with refractory painful distal symmetrical polyneuropathy, ultrasound-guided PRF of the stellate ganglion has been reported to show clinical benefit.
Lines 398-399: The efficacy of ultrasound-guided epidural injection for radiculopathy might be parallel to that achieved with conventional fluoroscopy-guided techniques. The ultrasound group may have the additional advantage of less inadvertent vascular puncture and shorter procedure time.
Line 402-403: Some studies suggest it may offer similar efficacy to transforaminal epidural injection in relieving pain and improving function in patients with cervical radiculopathy.
Line 426: The effectiveness of ultrasound- vs. fluoroscopy-guided caudal epidural injections for lumbar radicular pain has been evaluated in several studies and implied comparable outcomes.
Line 481: Ultrasound guidance may offer certain several benefits over fluoroscopy, though both techniques have distinct roles.
Comment:
It may be that ultrasound-guidance does have a particular benefit beyond radiation safety aspects (exposure is low as it is with use of fluoroscopy) but this chapter does not prove that. As an experienced clinician, I also disagree strongly with the comments that it is more accurate than fluoroscopy. We are talking apples and oranges here, with different technologies for different purposes.
Response:
We appreciate the reviewer’s important clarification. We acknowledge that current evidence does not establish ultrasound guidance as more accurate than fluoroscopy, and our manuscript does not present data suggesting such superiority. In our discussion, we describe radiation avoidance and the ability to visualize vessels and soft tissues as potential advantages of ultrasound without attributing any advantage in terms of procedural accuracy. To address this point, we have added: “Ultrasound guidance may offer certain several benefits over fluoroscopy, though both techniques have distinct roles.” and “It should be emphasized that there is a lack of high-certainty evidence confirming equivalent efficacy of ultrasound versus fluoroscopy guidance, especially for spinal injections. Moreover, the accuracy of ultrasound-guided procedures is highly machine- and operator-dependent; it may take years of training before novice practitioners feel confident in these techniques.”
Additionally, our manuscript aims to highlight current evidence on ultrasound-guided interventions without diminishing the well-recognized value and clinical utility of fluoroscopy. Indeed, some of the studies cited employed dual guidance, and we have reiterated: “For greater precision and safety, a dual-guidance approach—utilizing both ultrasound and fluoroscopy—may often be advisable.”
Comment:
Some of the sections mix pathologies (eg Trigeminal Neuralgia section) and this is unhelpful.
Response:
We thank the reviewer for this helpful observation. To address it, we have corrected the structure by relocating the case report describing postherpetic neuralgia in the left supraorbital nerve distribution to the appropriate section.
Comment:
It is in complete contrast to the recent articles and editorial in BMJ Open which suggest that injection therapies per se do not have therapeutic value. This was not addressed and is of major concern. The danger is the encouragement of others to 'have a go' when evidence is poor or even lacking.
Response:
The network meta-analysis titled “Common interventional procedures for chronic non-cancer spine pain: a systematic review and network meta-analysis of randomized trials,” published in BMJ, concluded that interventional procedures for axial or radicular chronic non-cancer spine pain may provide little to no pain relief. Specifically, for chronic radicular pain, moderate-certainty evidence indicated that epidural injections of local anesthetics and steroids (WMD −0.49 (−1.54 to 0.55)) and radiofrequency treatment of the dorsal root ganglion (WMD 0.15 (−0.98 to 1.28)) probably result in little to no difference in pain relief. Similarly, the BMJ Rapid Recommendations clinical practice guideline advised against epidural injections for chronic axial and radicular back pain.
However, these conclusions have been questioned due to the substantial heterogeneity among the pooled studies and potential methodological concerns, including possible omissions and data extraction errors. The meta-analysis combined diverse spinal regions, patient populations, diagnoses, and intervention types, undermining the clinical relevance and potentially leading to misleading conclusions. Such concerns were highlighted in the “Multisociety response to The BMJ publications on interventional spine procedures for chronic back and neck pain” editorial, authored by multiple international interventional spine societies.
We follow the reviewer’s suggestion and include the recent BMJ meta-analysis into our discussion section while emphasizing the importance of proper patient selection, informed consent and skilled practice as the following: “Nonetheless, a recent network meta-analysis reported minimal improvement by epidural injections or PRF for chronic radiculopathy.” and “These findings highlight the necessity of accurate diagnosis, careful patient selection, procedural expertise, and comprehensive informed consent regarding the potential benefits and limitations of injection-based therapies.”

Reviewer 2 Report
Comments and Suggestions for Authors
The article studies and discusses a highly relevant clinical topic, with the ambitious goal of providing an overview of ultrasound guidance in interventional procedures for the management of neuropathic pain. Overall, the article is well-structured and the topic is relevant (I find it very interesting and that there are gaps in the literature regarding awareness of the etiology of this symptomatic manifestation and a lack of consistency in terms of clinical treatments), but there are some aspects that, in my opinion, may deserve clarification.
In the introduction, the sentence "However, when there is disruption to the somatosensory pathway—whether due to trauma, infection, metabolic disorders, or neurodegenerative processes—aberrant signaling may occur" (lines 43-45) remains vague. It would be helpful to more clearly specify these mechanisms or conditions, even in an introductory context, so that it is effective and contributes well to the article's introduction. Furthermore, providing a single citation for such a broad and fundamental statement seems insufficient.
The authors identify their work as a pictorial narrative review in the title, but I believe it is important to clearly specify the specifics of this approach in the text. It is not sufficiently explained why this specific approach was chosen for the review. It would be helpful to briefly justify the rationale, for example, explaining why this format is suitable for the topic. At the same time, the authors should acknowledge the potential but also the intrinsic limitations of this approach, as this type of review could risk being more descriptive than critical. Clarifying this aspect would strengthen the article's external validity and help the reader better understand the framework adopted. A positive aspect is that the conclusions are consistent with the results presented: they are coherent, well aligned with the evidence discussed, and appropriately highlight the need for further, larger clinical studies.
The standard of English is good: the text is fluent, clear and appropriate to the scientific context.
Author Response
Reviewer 2
Comment:
The article studies and discusses a highly relevant clinical topic, with the ambitious goal of providing an overview of ultrasound guidance in interventional procedures for the management of neuropathic pain. Overall, the article is well-structured and the topic is relevant, but there are some aspects that, in my opinion, may deserve clarification.
Response:
We sincerely appreciate the reviewer’s positive feedback. The manuscript has been thoroughly revised in accordance with the reviewer’s suggestions.
Comment:
In the introduction, the sentence "However, when there is disruption to the somatosensory pathway—whether due to trauma, infection, metabolic disorders, or neurodegenerative processes—aberrant signaling may occur" (lines 43-45) remains vague. It would be helpful to more clearly specify these mechanisms or conditions, even in an introductory context, so that it is effective and contributes well to the article's introduction. Furthermore, providing a single citation for such a broad and fundamental statement seems insufficient.
Response:
To address the reviewer’s insightful comment, we have revised the introduction to provide greater clarity regarding the underlying mechanisms and conditions that contribute to aberrant somatosensory signaling. Specifically, we have expanded the description as follows: “Peripheral sensitization stems from alterations in nerve fiber density and ectopic hyperexcitability driven by changes in nerve membrane composition, synaptic properties, and neurotransmitter expression. These faulty discharges are subsequently propagated from peripheral neurons to their central targets. Central sensitization arises from synaptic changes in second-order neurons, microglial hyperactivation, dysfunction descending inhibition pathways and maladaptive plasticity in subcortical and cortical regions.” And “Based on anatomical classification, neuropathic pain can be categorized as central (e.g., stroke, spinal cord injury, multiple sclerosis, Parkinson's disease) or peripheral in origin (e.g., diabetes, radiculopathy, chemotherapy-induced peripheral neuropathy, acute inflammatory demyelinating polyneuropathy.)” Furthermore, multiple recent references were incorporated.
Comment:
The authors identify their work as a pictorial narrative review in the title, but I believe it is important to clearly specify the specifics of this approach in the text. It is not sufficiently explained why this specific approach was chosen for the review. It would be helpful to briefly justify the rationale, for example, explaining why this format is suitable for the topic. At the same time, the authors should acknowledge the potential but also the intrinsic limitations of this approach, as this type of review could risk being more descriptive than critical. Clarifying this aspect would strengthen the article's external validity and help the reader better understand the framework adopted.
Response:
We agree with the reviewer’s suggestion. The following explanation was added to justify the use of narrative review: “Ultrasound-guided interventions represent a new approach to treating neuropathic pain. Given the diversity of intervention techniques and study designs in this broad, evolving field, we elected to conduct a narrative review.”
We have acknowledged the inherent limitations of this approach in the discussion section: “Lastly, due to the inherent nature of narrative reviews, formal quality assessments of included studies were not performed, rendering the synthesis susceptible to subjective interpretation.”
Comment:
The conclusions are consistent with the results presented: they are coherent, well aligned with the evidence discussed, and appropriately highlight the need for further, larger clinical studies.
Response:
We thank the reviewer for the encouraging comment.

Reviewer 3 Report
Comments and Suggestions for Authors
The article is a continuation of the authors' previous work. It is richly illustrated and describes techniques for the use of ultrasound-guided neuralgia therapy. From a practitioner's point of view, this material is very valuable and addresses the topic of treating neuralgia using minimally invasive techniques, which have recently been gaining more and more supporters. It is worth noting that ultrasound itself, as a physical factor “per se,” used in neuralgia, has a beneficial effect.
The percent match should be lowered: 36%, which is probably the result of the use of standard medical descriptions and terms.
Author Response
Reviewer 3
Comment:
It is richly illustrated and describes techniques for the use of ultrasound-guided neuralgia therapy. From a practitioner's point of view, this material is very valuable and addresses the topic of treating neuralgia using minimally invasive techniques, which have recently been gaining more and more supporters.
Response:
We greatly appreciate the reviewer’s positive comment.
Comment:
It is worth noting that ultrasound itself, as a physical factor “per se,” used in neuralgia, has a beneficial effect.
Response:
The reviewer’s point is highly relevant, and we revised the final paragraph of the introduction to emphasize this point as follows: “Ultrasound is a widely utilized therapeutic and diagnostic tool. Therapeutic ultrasound has been shown to reduce neuropathic pain by modulating neurotransmission and inflammatory pathways, and it may even facilitate nerve regeneration. In diagnostic applications, the intricate architecture of peripheral and truncal nerves could be tracked effectively by high-resolution ultrasound.”
Comment:
The percent match should be lowered: 36%, which is probably the result of the use of standard medical descriptions and terms.
Response:
We appreciate the reviewer’s concern regarding text similarity. As this is a narrative review citing approximately 100 references, some similarity is expected due to the use of standard medical terminology and established descriptions. However, we confirm that the literature search, data analysis, and synthesis are original, and the manuscript was prepared entirely by the authors without plagiarism.

Reviewer 4 Report
Comments and Suggestions for Authors
The manuscript is well written and tells usefulness of ultrasound method during injections in each neuropathic pain. I consider that this article would be informative if a table which tells characteristics of each pain presents. Would you make a table which tells characteristics of pain, causes, part of injection, and effectiveness of injection and suggests how the ultrasound method is useful.
Author Response
Reviewer 4
Comment:
The manuscript is well written and tells usefulness of the ultrasound method during injections in each neuropathic pain.
Response:
We are grateful for the reviewer’s recognition.
Comment:
I consider that this article would be informative if a table which tells characteristics of each pain presents. Would you make a table which tells characteristics of pain, causes, part of injection, and effectiveness of injection and suggests how the ultrasound method is useful?
Response:
We sincerely thank the reviewer for this constructive suggestion. In response, we have created a summary table to provide readers with a concise reference for practical application. The table is presented as the following:

Reviewer 5 Report
Comments and Suggestions for Authors
The manuscript entitled "Ultrasound-Guided Interventions for Neuropathic Pain: a Narrative Pictorial Review" has been reviewed. While the topic is clinically relevant and of interest to the readership, the manuscript suffers from shortcomings in scientific rigor, novelty, and methodology.
The manuscript does not clearly establish its novelty compared to existing systematic reviews and narrative reviews on ultrasound-guided pain interventions. Without a clear contribution beyond pictorial illustrations, the added value is minimal.
The review does not contextualize findings against clinical practice guidelines in a critical manner; instead, it largely lists techniques.
Language and style are generally clear but often descriptive without critical synthesis.
The abstract promises a “comprehensive update,” but the content does not meet this expectation given methodological weaknesses.
Author Response
Reviewer 5
Comment:
While the topic is clinically relevant and of interest to the readership, the manuscript suffers from shortcomings in scientific rigor, novelty, and methodology. The manuscript does not clearly establish its novelty compared to existing systematic reviews and narrative reviews on ultrasound-guided pain interventions. Without a clear contribution beyond pictorial illustrations, the added value is minimal.
Response:
We sincerely thank the reviewer for this constructive critique, which has helped us strengthen the manuscript. In the revised version, we clarified the rationale for employing a narrative review format in the introduction: “Ultrasound-guided interventions represent a new approach to treating neuropathic pain. Given the diversity of intervention techniques and study designs in this broad, evolving field, we elected to conduct a narrative review.” Furthermore, we emphasized the novelty of this review by stating: “Most previous reviews have either compiled interventions by various imaging modalities or concentrated exclusively on specific pain conditions. To our knowledge, this is the first review specifically dedicated to ultrasound-guided interventions spanning the full range of neuropathic pain etiologies.”
Comment:
The review does not contextualize findings against clinical practice guidelines in a critical manner; instead, it largely lists techniques.
Response:
We appreciate this important observation. We have cited current guidelines in the second paragraph of the discussion and to compare them with our review, we added “As for interventional therapies the most consistent evidence to date pertains to her-pes-related pain and radiculopathy, which is consistent with our synthesis.” This underscores how the review not only summarizes existing techniques but also aligns them with guideline-based practice. Beyond existing guidelines, our review contributes additional value by providing pictorial demonstrations that are not typically included in guideline documents. In doing so, we hope to provide practical insights for clinicians while also facilitating further research.
Comment:
Language and style are generally clear but often descriptive without critical synthesis.
Response:
We acknowledge this limitation. In the revised manuscript, we explicitly recognized this shortcoming in the limitations section: “Lastly, due to the inherent nature of narrative reviews, formal quality assessments of included studies were not performed, rendering the synthesis susceptible to subjective interpretation.”
Comment:
The abstract promises a “comprehensive update,” but the content does not meet this expectation given methodological weaknesses.
Response:
We thank the reviewer for pointing this out. To avoid overstating the scope, we have removed the word “comprehensive.
Table 1. Summary of review findings
|
|
Pain characteristic |
Cause |
Patient positioning |
Ultrasound probe and technique |
Effectiveness of ultrasound-guided procedures |
Additional Notes |
|
Trigeminal neuralgia |
Paroxysmal facial pain in one or more divisions of the trigeminal nerve |
Neurovascular compression at root entry zone |
Head neutral
|
Linear probe, in-plane • V1: supraorbital notch • V2: infraorbital foramen • V3: mental foramen |
Sustained pain relief by nerve block (case study) |
Use Doppler imaging to avoid vascular puncture |
|
Lateral, with affected side up
|
Curvilinear probe, out-of-plane • Maxillary N.: pterygopalatine fossa • Mandibular N.: between pterygoid muscles |
|||||
|
Acute herpes pain |
Burning, stabbing, or itching pain in the affected dermatomal distribution |
Reactivation of VZV leading to viral nerve damage and inflammation |
Cervical: supine, head turned to contralateral side Thoracic: prone
|
Cervical: • SNRB • brachial plexus block • SGB Thoracic: • TPVB • ESPB |
Reduce pain and lower postherpetic neuralgia incidence by TPVB/ESPB (meta-analysis), PRF/SNRB/SGB (RCT) |
• Caution near large vessels (cervical) and lungs (thoracic) • Prefer non-particulate steroids |
|
Post-herpetic neuralgia |
Persistent pain after rash resolution |
Central and peripheral sensitization |
Effective pain control by PRF (RCT), TPVB/SNRB/brachial plexus block/SGB/PNS (case study) |
|||
|
Post-amputation pain |
Stump pain and phantom limb pain |
Multifactorial, including neuromas, nerve damage, sensitization |
• Variable depending on the target nerve • Neuroma identified as a hypoechoic, swollen mass adjacent to the parent nerve |
Significant pain reduction by PNS (RCT), nerve block/PRF (case study) |
Avoid direct injection into the nerve |
|
|
Painful polyneuropathy |
Numbness, tingling sensations, dysesthesia, weakness, balance impairment |
Metabolic, infections, connective tissue disorders, hereditary, toxins, nerve injuries, etc. |
• Variable depending on the target nerve • Commonly distal sensory nerves in lower extremities |
• Improvement of painful DPN by SGB (prospective cohort), lumbar plexus block/hydrodissection (case study) • Peripheral nerve injury pain decreased by nerve blocks/PNS (case study) |
Ultrasound facilitates targeting small distal nerves |
|
|
Painful radiculopathy |
Pain, numbness, clumsiness and even weakness in the distribution of the affected nerve root |
Mechanical compression and inflammatory irritation of the nerve root, often due to disc herniation or spondylosis |
Cervical: supine, head turned to contralateral side Lumbar: prone
|
Cervical: linear, in-plane • Transforaminal ESI • SNRB Lumbar: curvilinear probe for ESI/SNRB, linear probe for caudal block, in-plane • Transforaminal ESI • SNRB • Caudal block |
Improvement of pain by ESI (meta-analysis), SNRB/caudal block (RCT) |
• Caution in cervical region with large vessels • Prefer non-particulate steroids |
N.: nerve; VZV, varicella-zoster virus; SNRB: selective nerve root block; SGB: satellite ganglion block; TPVB, thoracic paravertebral block; ESPB, erector spinae plane block; PRF, pulsed radiofrequency; RCT, randomized controlled trial; PNS: peripheral nerve stimulation; DPN, diabetic polyneuropathy; ESI, epidural steroid injection.

Round 2
Reviewer 1 Report
Comments and Suggestions for Authors
This chapter suffers from over-reach and is written from the perspective of enthusiastic practitioners of ultrasound-guided blocks. It is not true that ultrasound is more accurate than fluoroscopically-guided injections and indeed this is admitted later in the chapter related to the epidural space. Each technology has a role. Several paragraphs have been added at the beginning related to mechanisms of action for controlling neuropathic pain - these are speculative with no mention of the important role that placebo likely plays. The section on trigeminal neuralgia remains a collection of different pathologies, and the section on post-amputation pain is highly misleading. The end of the chapter suddenly adopts a much more cautious tone, contradicting a lot of the earlier statements. As a pictorial review, there are too few pictures and far too much speculation.
Reviewer 5 Report
Comments and Suggestions for Authors
I am satisfied that you have adequately addressed the requested revisions.